# Insecticidal Activity of a Petroleum-Derived Spray Oil and an Organosilicone Surfactant on *Listronotus maculicollis* (Kirby) Adults in Laboratory and Greenhouse Bioassays

**DOI:** 10.3390/insects13111032

**Published:** 2022-11-08

**Authors:** Benjamin A. McGraw, Albrecht M. Koppenhöfer, Olga Kostromytska, Shaohui Wu, Steven R. Alm

**Affiliations:** 1Department of Plant Sciences, Pennsylvania State University, 243 ASI Building, University Park, PA 16802, USA; 2Department of Entomology, Rutgers University, 96 Lipman Dr., New Brunswick, NJ 08901, USA; 3Department of Plant Sciences and Entomology, University of Rhode Island, Kingston, RI 02881, USA

**Keywords:** annual bluegrass weevil, petroleum-derived spray oil, surfactant, turfgrass, resistance management

## Abstract

**Simple Summary:**

The annual bluegrass weevil (ABW), the most severe insect pest of short-mown turfgrass in eastern North America. Control programs traditionally target overwintering adults in spring (prior to egg laying) with broad-spectrum insecticides. However, the development of pyrethroid- and multiple-resistant populations has created the need for novel approaches. We investigated the insecticidal activity of a petroleum-derived spray oil and a surfactant on adults in laboratory and greenhouse trials. Both products caused rapid mortality, though were affected by carrier volumes, irrigation volume, and soil/substrate moisture. Neither products’ efficacy was affected by ABW pyrethroid resistance levels.

**Abstract:**

The annual bluegrass weevil (ABW), *Listronotus maculicollis* (Kirby), is a severe pest of golf course turf in eastern North America. The development of pyrethroid- and multiple-resistant populations has created a dire need for novel tactics to control adults. We examined the insecticidal properties of a petroleum-derived spray oil (PDSO; Civitas Turf Defense™.) and an organosilicone, nonionic soil surfactant (Silwet L-77^®^) in laboratory and greenhouse bioassays. Civitas and Silwet killed > 75% of ABW adults in multiple assays. The level of control was positively affected by increased rate, spray application volume, and soil moisture levels. Dissections of weevils treated with Civitas revealed material entering the insect’s hemocoel after 15–30 min, though most mortality occurred between the 3 and 24 h observation periods. Reducing rates while increasing carrier volume or soil moisture levels through irrigation applied prior to or after application also provided excellent control of adults in the same observation periods. Silwet provided comparable, yet less consistent levels of control in the laboratory studies but was excluded from further tests after treated plants demonstrated phytotoxicity in greenhouse studies. Neither Silwet nor Civitas efficacy was affected by pyrethroid resistance levels in the ABW populations tested.

## 1. Introduction

The annual bluegrass weevil (ABW), *Listronotus maculicollis* (Kirby) (Coleoptera: Curculionidae), is the most destructive insect pest of short-mown annual bluegrass (*Poa annua* L.) and creeping bentgrass (*Agrostis stolonifera* L.) in eastern North America [1]. The most important playing surfaces (e.g., greens, fairways, tee boxes) within this region are often compromised of one or both of these species. In spring, mated females will notch the stem of the turfgrass plant and place eggs into the oviposition scar, between the leaf sheath and the stem of the plant [2]. Upon eclosion, early instar larvae (1st–3rd) will tunnel the stem of the plant, occasionally exiting to attack a neighboring tiller. Later-instar (4th–5th) larvae feed externally on the plant and may sever stem at their base and cause death of apical meristems. Although the species is multivoltine, the spring generation is the more concentrated in space and typically the most damaging of the year. Turfgrass stands damaged during this period may fail to regenerate during the growing season, creating an inconsistent playing surface and prone to weed invasion.

Turfgrass managers attempt to limit ABW larval feeding through sequential insecticide applications between overwintering adult emergence in spring and completion of the third generation in late summer to early fall. Traditional approaches have relied upon broad-spectrum insecticides (e.g., pyrethroids, organophosphates) to control overwintered adults in spring prior to oviposition, since eggs are unaffected by most insecticides and only systemic insecticides affect the early instar larvae [3]. Adulticides are relatively inexpensive compared to larvicides, which has led to their overuse in time and space. Soil dwelling larvae represent the only other stages that can be effectively targeted with insecticides, though control requires precise timing to arrest crown feeding. The development of insecticide resistance has challenged both approaches. General enzymatic detoxification systems are at least partially responsible for the increased tolerance to insecticide exposure [4,5]. Additionally, ABW populations with elevated resistance levels may show a decrease in sensitivities to other insecticide classes (multiple resistance) such as those used in larval management [6,7,8].

Petroleum-derived spray oils (PDSOs) or horticultural oils have been used to control insects and fungi since the late 1800s, including horticultural pests and/or insect stages (e.g., eggs) that are typically not controlled with traditional synthetic chemistries [9]. Products may vary greatly based on their chemical structure and the distillation process, which in turn can affect their insecticidal properties [10,11] and potential for a phytotoxic response in the plant [12]. However, advances in refining technologies have led to more nuanced products for better integration into pest management programs. The mechanism(s) behind PDSO toxicity has been debated for decades [13], though recent work suggest that insect control may be achieved through several different modes of action including coating of the spiracles leading to anoxia [14], suffocation [15], desiccation through water loss [16], cell membrane disruption [17], or nervous system disruption [18]. Most PDSOs are mixed with soaps or nonionic surfactants that help the oil form a solution in water. These adjuvants may also possess insecticidal properties [19,20,21]. Despite possessing non-selective modes-of-action, PDSOs and surfactants are considered less harmful to non-target organisms [22] since they degrade rapidly [23]. PDSOs may also be integral components in managing insecticide resistant populations as several studies have documented synergistic activity with synthetic insecticides when applied together [24,25]. Furthermore, there have been no documented instances of pest populations developing resistance to PDSOs in more than a century of research.

The use of PDSOs for insect control in turfgrass has been largely unstudied, which in part may be due to the inability to achieve contact with subterranean pests and the potential for phytotoxicity. However, it stands to reason that several surface-active turfgrass insect pests (including ABW adults) may be susceptible to PDSO applications as they migrate through and on the turfgrass canopy. The objective of this study was to determine the insecticidal effects of Civitas Turf Defense™ (Suncor Energy Inc., Mississauga, ON, Canada) (hereafter referred to as ‘Civitas’), a PDSO currently marketed for disease and insect suppression in turfgrass. We also compared the insecticidal activity with an organosilicone, nonionic soil surfactant (Silwet^®^ L-77, Helena Chemical Company; Colliersville, TN, USA) (hereafter referred to as ‘Silwet’) with known toxicity to insects and mites [15,20]. Both Civitas and Silwet were evaluated against pyrethroid-susceptible and -resistant weevil populations to determine whether these products could be used in ABW resistance management programs.

## 2. Materials and Methods

### 2.1. Insects and Tests Materials

Weevils were collected from overwintering sites (e.g., wood lines, leaf litter) on golf courses in central Pennsylvania (Lewistown Country Club (LCC), Lewistown, PA, USA; laboratory experiments), central New Jersey (Pine Brook Golf Course (PB), Manalapan, NJ, USA; laboratory and greenhouse experiments) and northern New Jersey (Edgewood Country Club (EW), River Vale, NJ, USA, laboratory and greenhouse experiments) in late October using the extraction methods described in McGraw et al. (2011) [26]. Insects were taken to the laboratory where they were sorted, placed into unsexed batches of 50 to 100 into 840-mL plastic containers filled with moist (5% *v*/*w*) sand lining the bottom. The sand had been air-dried and pasteurized (3 h at 72 °C) before use. Containers were placed into an incubator (14 h light at 21 °C: 10 h dark at 14 °C). Adults were provided with black cutworm, *Agrotis ipsilon* (Hufnagel), diet (Bio-Serv, Frenchtown, NJ, USA) and supplemented *P. annua* clippings during their incubation. Adults that were not used within a few weeks were cold-stored for up to 2 months in an incubator (10 h light at 6 °C: 14 h dark at 4 °C) until they were warmed up for use in bioassay. Pyrethroid resistance bioassays were performed on a subset of the collected weevils using the methods described in Kostromytska et al. (2018) [27] The LCC weevils used in preliminary laboratory experiments assessing Civitas rate and spray volume displayed sensitivity (100% control) to bifenthrin at 0.5× field rates and could be considered “pyrethroid susceptible. Resistance levels of populations used in the remaining laboratory studies and all greenhouse studies were determined in previously published studies with adults [6,27]. LD50 values were obtained for bifenthrin in topical assays and compared to a susceptible population (Rutgers Horticultural Farm No. 2; North Brunswick, NJ, USA). Populations were categorized as susceptible (RR50 2.0: PB) and resistant (RR50 95: EW).

Civitas Turf Defense (98% mineral oil) and Silwet^®^ L-77 (99.5% organosilicone) were obtained from Suncor Energy Inc. (Mississauga, ON, Canada) and Helena Chemical Company (Colliersville, TN, USA), respectively.

### 2.2. Laboratory Bioassays

#### 2.2.1. Effect of PDSO Rate and Carrier Volume

Two experiments were conducted to determine the effect of Civitas rate and spray application or carrier (i.e., water) volume on adult mortality and speed of kill. Petri dishes (9 cm diameter × 1.5 cm height) were lined with a single filter paper of matching diameter. Each treatment within an experiment was replicated three times and each experiment was conducted three times.

In the first experiment, hand-pump atomizers were used to deliver one of five rates (0, 5.4 13.5, 27.1, and 54.1 L/ha) in a water carrier volume of 815 L/ha. Each treatment replicate consisted of a dish of 10 adult insects. After application, insects were provided *P. annua* leaves to feed upon. Mortality was assessed at 0.25, 0.5, 1, 3, 24, 48, and 72 h after application by removing the lid of the Petri dish and probing adults with forceps.

In the second experiment, we assessed the effect of one rate (13.5 L/ha) in carrier volumes of 815, 1630, 2445, and 3260 L/ha. These treatments were compared to water checks applied in the same carrier volumes at 0.25, 0.5, 1, 3, 24, 48, and 72 h after application. Adults killed in bioassays were removed from dishes, rinsed with water, and placed in an −80 °C freezer until dissections could be made to determine the route of entry of Civitas into the body. Civitas is a mixture of food-grade paraffins (alkanes) and emulsifiers, which is then mixed with a pigment (“Harmonizer”) to reduce potential for phytotoxicity. The Harmonizer makes identification of the material on and within the insect possible with a dissecting microscope by observing the process of pigment penetration through the insect cuticle and diffusion inside the insect hemocoel and internal organs at different times after treatment. Dissections were performed using the methodologies described in McGraw et al. (2020) [28].

#### 2.2.2. Effect of PDSO and Soil Surfactant Rate and Carrier Volume

An experiment was conducted to determine the effect of carrier volume of PDSO and soil surfactant on adult mortality. It used similar methodologies to those in Section 2.2.1 except for adding two layers of pre-moistened filter papers per Petri dish (to remove the potential for pooling of excessive moisture and possible drowning) and the inclusion of a soil surfactant (Silwet^®^, Helena Chemical Company, Collierville, TN USA). Filter papers were pre-moistened with 1.75, 1.5, and 1.0 mL of water, so the final water content in each dish after spraying treatments was 2 mL so that the papers were saturated but free of excess water. Ten unsexed adults were introduced in the dishes with pre-moistened filter paper. Three rates of Civitas (8.0, 12.7, and 19.1 L/ha) and Silwet (0.56, 0.80, and 1.27 L/ha) were sprayed at three different carrier volumes (408, 815, and 1630 L/ha = 0.25, 0.5, and 1 mL/dish) using 100-mL atomizers. Control dishes were sprayed with water at the same three volumes. The experiment was conducted three times with two Petri dishes per treatment per experimental run. Mortality was evaluated at 3, 24, 48, and 72 h after treatment application.

#### 2.2.3. Effect of Substrate Moisture on Mortality

Similar methodologies as in Section 2.2.2 were used to assess the effect of substrate moisture on mortality. Only one rate of each product was tested (Civitas: 9.6 L/ha; Silwet: 0.56 L/ha) based on the results from Section 2.2.2. Moisture in the filter paper-lined Petri dishes was added to reach the final moisture level of 50, 100, 150% saturation (1, 2, 3 mL per Petri dish including spray volume). Treatments were sprayed at a carrier volume of 815 L/ha; controls were sprayed with water only. This experiment was conducted four times with two Petri dishes per treatment per experimental run.

#### 2.2.4. Effect of Pyrethroid Resistance on Mortality

The susceptibility to Civitas and Silwet of a bifenthrin-susceptible (PB) and a bifenthrin-resistant population (EW) was tested using the same methodologies described in Section 2.2.2 and Section 2.2.3. Civitas (9.6 L/ha) and Silwet (0.8 L/ha) were sprayed at a carrier volume of 815 L/ha. Mortality was assessed at 3, 24, 48 and 72 h after treatment application. Three Petri dishes per treatment were used in each of three experimental runs.

### 2.3. Greenhouse Studies

A series of greenhouse experiments were conducted to study the effect of rate, carrier volume, soil moisture, and post-application irrigation in more complex or field-like conditions. The turfgrass used in experiments was grown in the greenhouse on a mix of pasteurized (3 h at 70 °C) sandy loam (61% sand, 27% silt, 12% clay; 2.3% organic matter, pH 5.9) and pasteurized play sand (3:1 ratio) in 540-mL deli-cups (Fabri-Kal^®^, Kalamazoo, MI, USA) with drainage holes. Creeping bentgrass (cv. Penncross) was seeded directly into the pots at 6.25g m^−2^ and grown in the greenhouse for 1 months prior to use. Plants were fertilized weekly (20-20-20 NPK, The Scotts Miracle Gro Co., Marysville, OH, USA), watered as necessary and clipped twice a week to maintain 1.3 cm height. In each experiment, 10 unsexed weevils were introduced into each pot 3 h before treatment application and allowed to acclimate. Adults from the PB population were introduced to half of the pots (two pots) and EW adults to the other half (two pots) in each experimental run. Pots with adults were covered with screened lids before and after treatments were applied. Treatments were applied using a Generation III Research track sprayer (Devries Manufacturing, Hollandale, MN, USA) that allowed for controlling spray rates and carrier volumes. Experiments were evaluated 3 days after treatment (DAT) by submerging pots into warm water to extract adults (overall 97–98% recovery). Missing weevils were counted as dead.

#### 2.3.1. Effect of Carrier Volume and Rate

The soil in the pots was saturated by irrigation (to 100% saturation) prior to treatment applications. No irrigation was provided after treatment application. Control pots received water at the same carrier volumes as used for the treatments. In the first experiment, treatments consisted of four rates of Civitas (12.7, 25.5, 38.2 and 54.1 L/ha) or three of Silwet (1.6, 3.2, or 6.4 L/ha), all applied in carrier volumes of either 815 or 1630 L/ha. Because there was little mortality and severe phytotoxicity in the Silwet treatments, in the second experiment only Civitas was tested at one rate (54.1 L/ha) delivered in one of three carrier volumes (815, 1630 or 3.260 L/ha). The experiments were conducted in three runs with four replicates per run (two pots with population EW, two pots with population PB).

#### 2.3.2. Effect of Soil Moisture on PDSO Efficacy

The same methodologies were employed as in Section 2.3.1, but soils were weighed prior to providing irrigation, so that soil moisture could be manipulated to achieve three levels of saturation (100%, 75% and 50%). After pre-application irrigation was applied, Civitas was delivered at the rate of 54.1 L/ha in a carrier volume of 1630 L/ha. Control pots were treated with water only. The experiment was conducted in three runs with four replications per treatment each (two pots with EW population, two pots with PB population).

#### 2.3.3. Effect of Post–Treatment Irrigation on PDSO Efficacy

The same methodologies as in Section 2.3.1 were used to assess the impact of post application irrigation on mortality. Soils were saturated by irrigation (to 100% saturation) prior to application. Civitas (54.1 L/ha) was applied in one of two carrier volumes (815 or 1630 L/ha). Then, pots were irrigated with 0, 1.27 or 2.5 mm of water. Control pots were sprayed with water only. Two experimental runs with four replications per treatment each (two pots with EW population, two pots with PB population) were arranged for this experiment.

### 2.4. Statistical Analysis

Statistical analyses were performed using Statistix 9.0 software package (Tallahassee, FL, USA) and SAS 9.3 (Cary, NC, USA). Normality of datasets was assessed by Shapiro–Wilk tests and examining Q-Q plots. The proportion of adults controlled in the laboratory experiments in Section 2.2.1 was control corrected before being arcsine transformed to meet assumptions of normality. Mortality in the untreated controls were low in both laboratory trials (average < 10%) and therefore control corrections were conducted using the untreated mortality within the time period (first laboratory experiment) or untreated with the same spray volume as the treatment within the time period (second laboratory experiment). Polynomial contrasts were constructed to assess linear, quadratic, and cubic relationships between increasing treatment rates or spray volume and ABW mortality. One-way ANOVA was performed to separate treatment means within time periods. In greenhouse experiments, adult counts within each experimental repetition were control corrected before analysis. Percent control data were normalized by arcsine square root transformation or square root transformation prior to analysis of variance. Means were separated using Tukey’s HSD test (α = 0.05). In Greenhouse experiments 1–3, the effect of Civitas and Silwet (treatment), the rate of the application and (rate), and application volume on percent of control was determined by conducting analysis of variance (GLM procedure, SAS institute). For all experiments, if mortality was observed in the control, mortality was control corrected (Abbott, 1925).

## 3. Results

### 3.1. Laboratory Studies

#### 3.1.1. Effect of PDSO Rate and Carrier Volume

A significant linear relationship was observed between insect mortality and Civitas rate (F = 26.75; df = 3, 251; *p* < 0.0001) and time (F = 7.04; df = 5; 251; *p* < 0.0001) in the first laboratory experiment. Mortality in the untreated controls at 72 h (10%) and subsequent control correction caused for a significant cubic relationship to be found for time as well (*p* = 0.0002). When data were analyzed within time period, significant differences between treatments were observed between the high rate of Civitas (54.1 L/ha) and lower rates as early as 30 min (F = 4.97; df = 3, 35; *p* = 0.006) through 24 hrs (F > 8.4; df = 3, 35; *p* < 0.001). The highest rate caused significantly higher mortality (78–80% at 72 h) than all other treatments at all time periods (F ≥ 4.97; df = 3, 35; *p* ≤ 0.006), with the exception of the 15 min mark (Figure 1 top).

In the second experiment with varying application volumes, insect mortality was significantly affected by spray volume (F = 37.69; df = 3, 248; *p* < 0.0001) and time (F = 51.13 df = 5, 248; *p* < 0.0001). Similar to the first experiment, no mortality was observed with 13.5 L/ha applied in 815 L/ha between 0 and 1 hr (Figure 1 bottom). Higher spray volumes (2445–3260 L/ha) significantly improved control (F = 11.64; df = 3, 35; *p* < 0.001) as early as 1 hr (7–20%). Spray volumes equal or greater than 1630 L/ha significantly improved mortality at 24 (52–73%) and 72 h (61–73%) compared to the standard 815 L/ha carrier (F > 5.16; df = 3, 35; *p* < 0.005).

Dissections of killed weevils revealed that Civitas entered the body of the insects between the 15- and 30 min observation periods. The green pigment adjuvant in the PDSO was observed within the trachea, digestive system, and in the hemocoel opposite of spiracles in the abdomen, suggesting that the material did not move great distances into the body, but entered through multiple routes.

#### 3.1.2. Effect of PDSO and Soil Surfactant Rate and Carrier Volume

For Civitas, mortality significantly increased with time (F = 3.86; df = 3,66; *p* = 0.01) but most of the mortality occurred within 24 h for both treatments with no significant increases thereafter (Figure 2). Rate was a significant factor with mortality generally being higher at 12.7 L/ha and 19.1 L/ha than at 8.0 L/ha at all observations intervals (F ≥ 11.29; df = 3, 53; *p* < 0.01). Mortality was also affected by carrier volume generally being higher at 408 L/ha and 815 L/ha than at 1630 L/ha at all observation intervals (F ≥ 8.54; df = 3, 53; *p* < 0.01). There was no significant rate * volume interaction at any observation interval.

For Silwet, mortality did not increase with time (F = 1.05; df = 3.66; *p* = 0.37) (Figure 2). Mortality was significantly affected by rate at all observation intervals (F ≥ 15.98; df = 2, 53; *p* < 0.01) but was not affected by carrier volume at any interval (F ≤ 1.20; df = 2, 53; *p* ≥ 0.31). There was no significant rate * volume interaction at any observation interval. In the Petri dishes treated with Silwet, partial recovery of initially knocked down weevil was observed.

#### 3.1.3. Effect of Substrate Moisture on Mortality

Moisture level significantly affected mortality at each evaluation interval for Civitas (F ≥ 37.48; df = 2, 53; *p* < 0.01) and Silwet (F ≥ 15.19; df = 2, 53; *p* < 0.01) (Figure 3). For both products, very limited mortality (0–4%) was observed at 50% moisture, which was significantly lower than for the higher moisture levels. For Civitas, mortality at 150% moisture level was significantly higher than at 100%. For Silwet, mortality was numerically higher by 24–30% at 150% than at 100% at the various observation intervals, but there was no statistically significant difference between these moisture levels.

#### 3.1.4. Effect of Pyrethroid Resistance on Mortality

No significant differences were detected at any observation interval between pyrethroid resistant and susceptible ABW populations in their susceptibility to Silwet (F = 4.81; df = 3.95; *p* < 0.01) and Civitas (F = 0.10; df = 3.144; *p* = 0.78) (Table 1).

### 3.2. Greenhouse Studies

#### 3.2.1. Effect of Carrier Volume and Rate

In the second experiment, Civitas (at a constant rate of 54.1 L/ha) caused significant mortality (F = 210.88; df = 1.71; *p* < 0.001) but carrier volume had a marginal effect on ABW mortality (F= 2.7; df = 2.71; *p* = 0.07); both factors marginally interacted (F = 2.74; df = 2.71; *p* = 0.07). Mortality did not differ between the pyrethroid-susceptible and –resistant populations (*p* = 0.11). Civitas caused significantly higher mortality compared to the untreated control pots with the same application volume, and mortality was significantly higher at the highest carrier volume (3260 L/ha) than at the lowest (815 L/ha) (Figure 4).

#### 3.2.2. Effect of Soil Moisture on PDSO Efficacy

Mortality was significantly affected by Civitas treatment (F = 160.7; df = 1.71; *p* < 0.01) and soil moisture level (F = 160.7; df = 1.71; *p* < 0.01) but not by pyrethroid-susceptibility of the population (F = 0.17; df = 1.71; *p* = 0.76). Moisture and treatment interacted significantly (F = 7.6; df = 2.71; *p* < 0.01). Mortality in the pots not treated with Civitas was minimal (<1%) irrespective of soil moisture level. In the Civitas treated pots, mortality was significantly higher at 100% soil moisture (63%) than at 50% (30%) and 75% moisture (43%) (Figure 5).

#### 3.2.3. Effect of Post–Treatment Irrigation on PDSO Efficacy

Significantly higher mortality was observed in Civitas- treated pots (irrespective of carrier volume and post application irrigation treatment) than in untreated controls (F = 284.9; df = 1.95; *p* < 0.01). No significant differences in mortality were detected between Civitas carrier volumes at any post application irrigation treatment (F = 0.56; df = 1.95; *p* = 0.45). Combining volumes revealed a significant effect of post-application irrigation on Civitas efficacy when compared within Civitas treatments (F = 7.1; df = 2.95; *p* < 0.01) or when compared to that of no-irrigation treatments and the untreated controls (Figure 6). No differences in mortality were detected between Civitas treatments followed by 1.27 or 2.5 mm of post application irrigation.

## 4. Discussion

Our findings from both laboratory and greenhouse studies suggest that Civitas can be highly effective in controlling ABW adults, though mortality is dependent on rate, carrier volume and/or soil moisture. Increasing Civitas rates led to increased mortality, with most mortality occurring between 3 and 24 h. Reducing rates while increasing carrier volume or soil moisture levels through irrigation (applied prior to or after application) also provided >80% control of adults in the same time periods. Silwet provided comparable, yet less consistent levels of control in the laboratory studies. Neither Silwet nor Civitas applications were affected by the pyrethroid resistance level of the tested ABW populations. Silwet was removed from further examination when greenhouse plants treated with rates required to provide acceptable ABW control demonstrated phytotoxic responses. Combined, these findings suggest that Civitas has potential as a replacement for organophosphates and pyrethroids in ABW adult management, especially for populations with elevated pyrethroid resistance levels.

Civitas-induced mortality started within 30 min but increased until 24 h after application. Historically, PDSOs have been assumed to work primarily by clogging spiracles leading to anoxia [14]. With heavy saturated oils (>nC23) this may be the case, as these materials coat the insect’s cuticle and penetrate only short distances into the body [16]. Over the last 40 years, several other mechanisms have been elucidated (e.g., cell membrane and nervous system disruption) with light, unsaturated hydrocarbon oils (<nC19). Civitas is a mixture of isoparaffins which range between 16 and 33 carbons mixed with an emulsifier [29] and therefore, its mode of action on insects cannot be determined by structure alone. We observed the green phthalocyanine pigment or Harmonizer component of Civitas in the trachea adjacent to the spiracles, the hemocoel around fat bodies, and the gut, suggesting that the material entered through multiple routes. Given that most mortality occurred between 3 and 24 h, it is probable that Civitas causes death through anoxia which is likely to result after hours of exposure to PDSOs [30].

Our laboratory and greenhouse observations suggest that moisture is critical to the insecticidal properties of both Silwet and Civitas. However, satisfying the need for high moisture in field situations may be prohibitive for some operations. Saturating turfgrass soils is not desirable as the increased moisture softens and decreases the playability of the surface, and may lead to biotic (e.g., disease) or abiotic (e.g., rutting from equipment tires) damage. Spring applications targeting migrating overwintering adults may benefit from applying post-application irrigation or by making applications in advance of rainfall. However, control may be compromised if rainfall does not adequately move the material to the target and/or the material dries on the foliage before irrigation is applied or rainfall occurs. Additionally, achieving adequate control through the use of high carrier volumes may also reduce practitioner adoption due to the increased time required to apply the material. The average 18-hole US golf course has 12.1 ha of fairways, 1.2 ha of tees, and 0.43 ha of greens/collars [31]. Typical sprayer tank volumes range between 662 and 1136 L which would require many hours to days to treat just the most valued areas (e.g., putting greens) at the highest spray volumes examined in this study (2445 and 3260 L/ha).

Increasing the Civitas rate while maintaining a standard carrier volume (e.g., 407.5–814.9 L/ha) and applying moderate amounts (1.27 or 2.5 mm) post-application irrigation may prove to be the most practical means to using PDSOs to control ABW. Unlike for Silwet, phytotoxicity was not observed with increasing rates of Civitas (up to 54.1 L/ha). This may be due in part to the Harmonizer (pigment) masking phytotoxicity. Kreuser and Rossi (2014) demonstrated that bi-monthly applications of Civitas alone (50 L/ha), but not Harmonizer (3 L/ha) resulted in chlorosis in creeping bentgrass [32]. Growth chamber studies suggested that the phytotoxic response was in part due to oil residue accumulation on the plant. Turfgrass managers attempting to control ABW with Civitas may wish to limit high-rate applications or increase the interval between applications. ABW adults may emerge from overwintering in spring over several weeks [28,33], which may often result in multiple insecticide applications to reduce oviposition and future larval damage if not well-timed. Applications of high rates of Civitas in short time frames (<2 weeks) could lead to declining turfgrass health, as the Harmonizer alone has been documented to lead to decreases in stand density over time [32].

We did not observe a difference in PDSO or surfactant insecticidal activity between pyrethroid-resistant (RR = 95) and susceptible populations. Turfgrass managers who manage pyrethroid-resistant populations report the need for greater than average number of insecticides required throughout the growing season [3]. Both cross resistance to pyrethroids and multiple resistance (to both organophosphates and several larvicide classes) has been documented in populations with elevated pyrethroid resistance [7,8]. Optimizing oil-based IPM programs for ABW management would benefit both managers of resistant and susceptible populations, as currently the only insecticide options for controlling adults are pyrethroids and organophosphates. PDSOs would greatly assist in delaying the development of resistance in susceptible populations and help to replace broad-spectrum insecticides in favor of products with improved eco-toxicological profiles. PDSO modes of action, lack of residual activity, and low probability of resistance development also make them sustainable management tools. However, more research is needed to optimize adult control in the field while minimizing water use and maximizing plant health.

## Figures and Tables

**Figure 1 insects-13-01032-f001:**
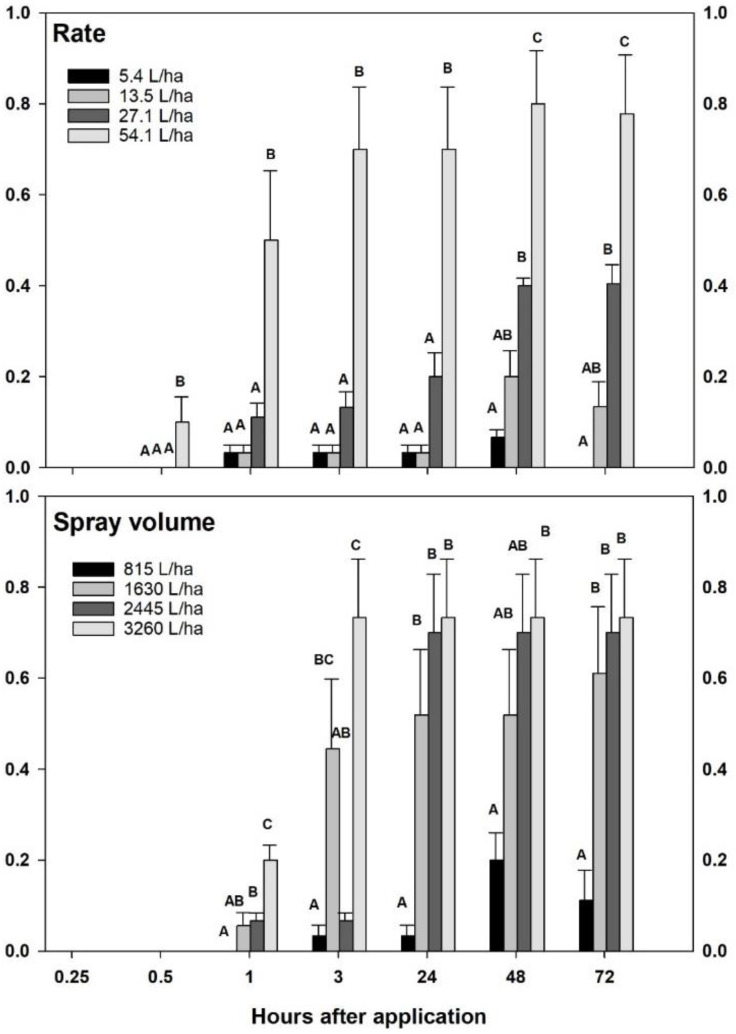
Cumulative effect of Civitas rate (top panel; at constant 815 L/ha carrier volume) and carrier volume (bottom panel; at constant 13.5 L/ha Civitas) on adult *Listronotus maculicollis* mortality between 0.25 to 72 h after application. Treatments within each interval were control corrected prior to analysis. Where polynomial contrasts detected significant relationships, one-way ANOVA was used to separate treatment means within time intervals at the at α= 0.05. Means marked with the same letter do not differ significantly from one another.

**Figure 2 insects-13-01032-f002:**
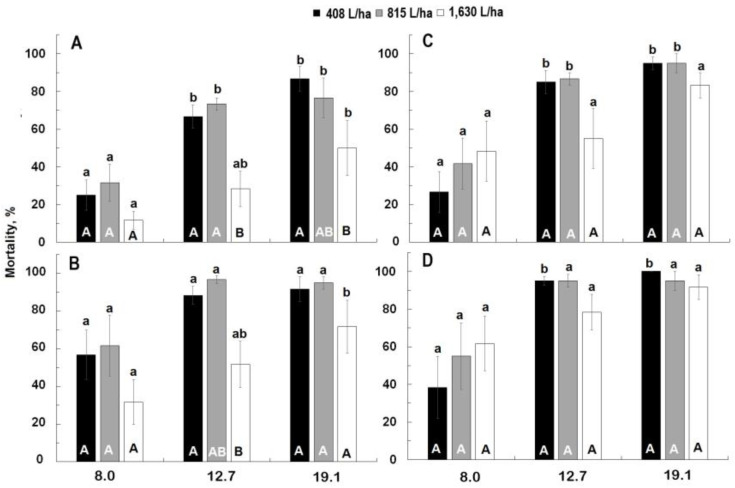
Effect of three Civitas rates ((**A**,**B**): 8.0, 12.7, 19.1 L/ha) and three Silwet rates ((**C**,**D**): 0.56, 0.80, 1.27 L/ha) applied at different carrier volumes (408, 815, 1630 L/ha) on mortality of adult *Listronotus maculicollis* evaluated at 3 h (**A**,**C**) and 24 h (**B**,**D**). Means marked with the same lower-case letter do not differ statistically within the application volume. Means marked with the same capital letter do not differ significantly within rate.

**Figure 3 insects-13-01032-f003:**
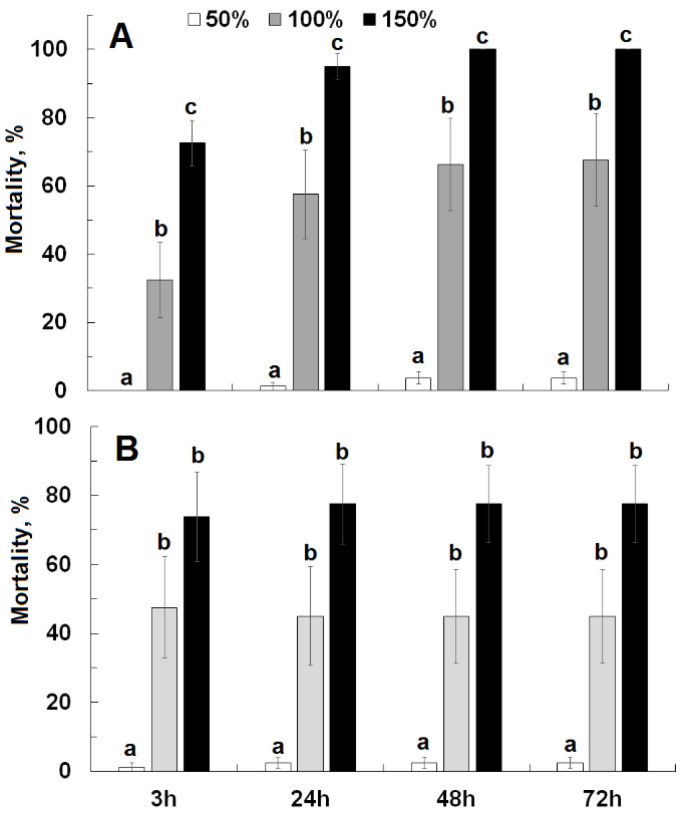
Cumulative effect of Civitas ((**A**); 9.6 L/ha) and Silwet ((**B**); 0.56 L/ha) applied at different moisture levels (50%, 100%, 150% filter paper saturation) on mortality of *Listronotus maculicollis* adults evaluated at 3, 24, 48 and 72 h after treatment. Means marked with the same letter do not differ statistically within each evaluation time.

**Figure 4 insects-13-01032-f004:**
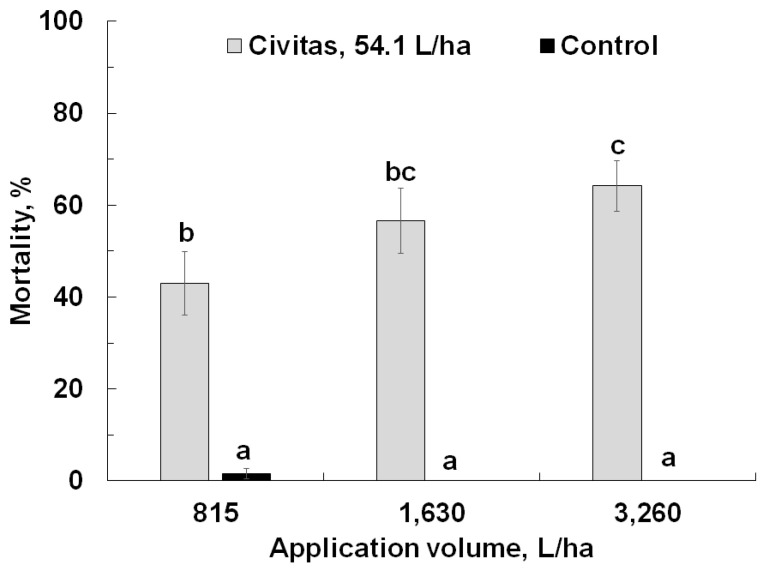
Effect of Civitas (54.1 L/ha) on mortality of *Listronotus maculicollis* adults at different carrier volumes. Means marked with the same letter do not differ statistically.

**Figure 5 insects-13-01032-f005:**
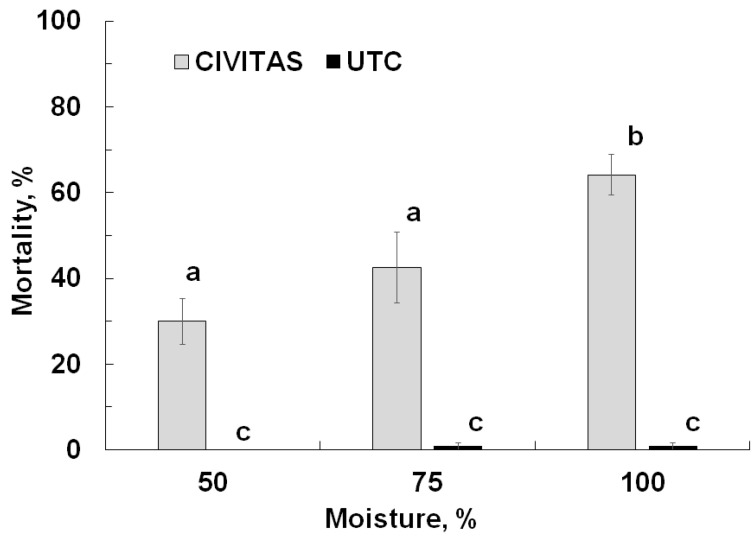
Mortality of *Listronotus maculicollis* (not corrected) adults caused by Civitas (54.1 L/ha applied in 1630 L/ha carrier volume) at different soil moisture levels. Means marked with the same letter do not differ statistically.

**Figure 6 insects-13-01032-f006:**
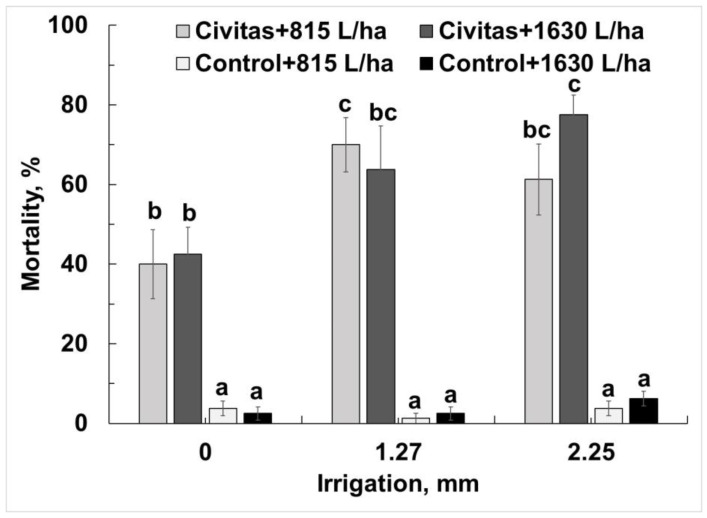
Effect of post treatment irrigation on Civitas (51.4 L/ha) efficacy against *Listronotus maculicollis* adults (data combined across different carrier volumes). Means marked with the same letter do not differ statistically within each evaluation time. Carrier volume (815 or 1630 L/ha) had no effect on ABW mortality.

**Table 1 insects-13-01032-t001:** Percent mortality of susceptible (PB) and resistant (EW) *Listronotus maculicollis* adults exposed to Civitas (9.6 L/ha) and Silwet (0.8 L/ha) (815 L/ha carrier volume, 100% filter paper saturation).

Treatment	Population	3 h	24 h	48 h	72 h
Civitas	EW	33.3 ± 11.4 a ^1,2^	63.3 ± 12.0 a ^1,2^	73.3 ± 9.4 a ^1,3^	74.4 ± 8.9 a ^1,3^
	PB	44.4 ± 13.0 a	70.0 ± 14.7 a	72.2 ± 14.3 a	72.2 ± 14.3 a
Silwet	EW	71.1 ± 11.7 a	64.4 ± 11.4 a	61.1 ± 11.2 a	60.0 ± 11.2 a
	PB	73.3 ± 9.6 a	68.9 ± 10.1 a	64.4± 10.7 a	60.0 ± 10.4 a
UTC	EW	0 b	0 b	1.1 ± 1.1 b	2.2 ± 1.5 b
	PB	0 b	0 b	1.1 ± 1.1 b	3.3 ± 1.7 b

^1^ The difference among between the resistant and susceptible population were not significant at α =0.05 (F ≤ 0.35; df = 1, 53; *p* ≥ 0.56). ^2^ Data were not control corrected. ^3^ Data were corrected for control mortality. Means followed by the same lowercase letter do not differ significantly from one another within observation period.

## Data Availability

The data presented in this study are available on request from the corresponding author.

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
