# Peer review of "Insecticidal Activity of a Petroleum-Derived Spray Oil and an Organosilicone Surfactant on Listronotus maculicollis (Kirby) Adults in Laboratory and Greenhouse Bioassays"

_insects, 2022, doi:10.3390/insects13111032_

Round 1

Reviewer 1 Report

This is a well-written manuscript with important findings regarding the insecticide resistance management of the annual bluegrass weevil. I recommend it for publication after a few minor edits provided in the attached file.

Author Response

Line 21: Italicized

Line 111: Italicized

Line 120: Placed numbers in [ ]

Line 157, 160(2),161, 162: Included “and”

Line 257: Changed to “The highest rate caused significantly higher mortality (78-80% at 72 hrs) than all other treatments at all time periods (F > 4.97; df = 3, 35; p < 0.006), with the exception of the 15 min mark (Figure 1 top).

Line 285: Changed to observation

Line 318: Italicized

Reviewer 2 Report

This manuscript by McGraw et al., explores the efficacy of two oils, Civitas and Silwet, at controlling the annual bluegrass weevil on turf. Overall this is a straightforward manuscript that demonstrates the impact of these oils. The authors conduct toxicity assays in both the lab on filter paper and in the greenhouse on small pots of turfgrass. In both cases treatment with various concentrations and delivery volumes significantly increased weevil mortality. They also demonstrated that these compounds were equally effective on pyrethroid resistant and susceptible strains. Interestingly, they also looked at the points of entry for Civitas using a green additive and found that while the compound did enter trachea, the gut and the hemocoel, the compound did not disseminate long distances in the insect. Overall, this is an interesting manuscript that demonstrates that an older control strategy (oils) can prove effective when resistance has developed against front-line insecticides. The studies are well executed and controlled and I only have a few minor comments.

In figure 2 there does not appear to be a statistical analysis for Silwet between rates (i.e. capital letters).  

For figures with time treatments (i.e., figures 1 and 3) are the mortality effects cumulative over time?

In figure 3B it seems odd that there was no increased mortality over time. Do the authors have an explanation for this? Error bars also seem remarkably consistent across treatments.

Overall, these concerns are minor and the manuscript, although of somewhat limited impact, will likely be of interest to readers of Insects. The authors also do a good job discussing the limitations of this approach (phytotoxicity, spray volume, etc.) in the discussion. I think with minor modifications this article will be appropriate for publication in Insects.

Author Response

In figure 2 there does not appear to be a statistical analysis for Silwet between rates (i.e. capital letters).  

Line 289: Added updated figure with letters to indicate means separation

For figures with time treatments (i.e., figures 1 and 3) are the mortality effects cumulative over time?

Cumulative. Added “Cumulative” to figure legends on Figures 1 and 3

In figure 3B it seems odd that there was no increased mortality over time. Do the authors have an explanation for this? Error bars also seem remarkably consistent across treatments.

Comments on Fig. 3.  Silwet simply works fast so that there is no additional mortality over time, just as there was not in Fig. 2.  The error bars are similar but not exactly the same.  One just has to increase the size of the figure or screen to see it.  I measured it.